# Observations of the downwelling far-infrared atmospheric emission at the Zugspitze observatory

Luca Palchetti[1], Marco Barucci[1], Claudio Belotti[1], Giovanni Bianchini[1], Bertrand Cluzet[4], Francesco D'Amato[1], Samuele Del Bianco[5], Gianluca Di Natale[1], Marco Gai[5], Dina Khordakova[3], Alessio Montori[1], Hilke Oetjen[7], Markus Rettinger[2], Christian Rolf[3], Dirk Schuettemeyer[7], Ralf Sussmann[2], Silvia Viciani[1], Hannes Vogelmann[2], and Frank Gunther Wienhold[6]

[1]National Institute of Optics - CNR-INO, Florence, Italy
[2]Karlsruhe Institute of Technology, IMK-IFU, Garmisch-Partenkirchen, Germany
[3]Forschungszentrum Jülich, Germany
[4]Univ. Grenoble Alpes, Université de Toulouse, Météo-France, CNRS, Centre d'Études de la Neige, Grenoble, France
[5]Institute of Applied Physics - CNR-IFAC, Florence, Italy
[6]Institute for Atmospheric and Climate Science - ETH, Zürich, Switzerland
[7]ESA/ESTEC, Noordwijk, Netherlands

**Correspondence:** Luca Palchetti, (luca.palchetti@cnr.it)

**Abstract.**

Measurements of the spectrum of the atmospheric emission in the far-infrared (FIR) range, between 100 and 667 $cm^{-1}$ (100–15 $\mu$m) are scarce because of the detection complexity and of the strong absorption of air at ground level, preventing the sounding of the FIR from low altitude. Consequently, FIR measurements need to be made from high-altitude sites or on board airborne platforms or satellites. This paper describes the dataset of FIR spectral radiances of the atmosphere and snow surface emission measured in the 100–1000 $cm^{-1}$ range by the Far-Infrared Radiation Mobile Observation System (FIRMOS) instrument during a 2-month campaign carried out from ground at about 3000 m of altitude on the top of Mount Zugspitze in the German Alps, in 2018-2019. This campaign is part of the preparatory activity of a new space FIR mission, named Far-infrared-Outgoing-Radiation Understanding and Monitoring (FORUM), which is under development by the European Space Agency (ESA). The dataset acquired during the campaign also includes all the additional measurements needed to provide a full characterisation of the observed atmospheric state and the local atmospheric and surface conditions. It includes co-located spectral measurements in the infrared range from 400 to 1800 $cm^{-1}$; lidar backscatter profiles; radiosoundings of temperature, humidity and aerosol backscatter profiles; local weather parameters; and snow/ice microphysical properties. These measurements provide a unique dataset that can be used to perform radiative closure experiments to improve modelling parameters in the FIR that are not well characterised, such as water vapour spectroscopy, scattering properties of cirrus clouds and the FIR emissivity of the surface covered by snow. The consolidated dataset is freely available via the ESA campaign dataset website at https://doi.org/10.5270/ESA-38034ee (Palchetti et al., 2020a).

## 1 Introduction

The Far-InfraRed (FIR), defined here as the longest wavelength region of the infrared spectrum covering the wavenumber range from 667 cm$^{-1}$ (or equivalently 15 $\mu$m wavelength) down to 100 cm$^{-1}$ (100 $\mu$m), contains more than 50% of the energy emitted by the Earth toward space. This spectral region is modulated by the properties of the most relevant components of the climate system, such as water vapour, carbon dioxide, clouds, and snow surface emissivity (Harries et al., 2008). The characterisation of the FIR radiative properties of these components is therefore essential to improve gaseous modelling of spectroscopy (Mlawer et al., 2019), cirrus cloud radiative properties (Cox et al., 2010) and surface emissivity (Chen et al., 2014), which in turn will allow a better understanding of the Earth radiation budget and to reduce uncertainties in climate models (Huang et al., 2018; Baran et al., 2014; Huang et al., 2007). Nevertheless, due to technical challenges, systematic and global measurements of the FIR from space are still missing, and are scarce from ground and airborne platforms.

To fill this observational gap, a space mission, named Far-infrared-Outgoing-Radiation Understanding and Monitoring (FO-RUM) (Palchetti et al., 2020b), is under development by the European Space Agency (ESA) as the ninth Earth Explorer mission to be launched in 2026 (https://www.forum-ee9.eu/). This mission will measure with high accuracy the spectrum of the outgoing infrared radiation from 100 to 1600 cm$^{-1}$ (100–6.25 $\mu$m) covering, for the first time with high spectral resolution, the FIR portion of the spectrum. In preparation for this mission, an instrument demonstrator, named Far-Infrared Radiation Mobile Observation System (FIRMOS), was developed for field applications from ground-based (and in perspective airborne) platforms to verify with real measurements the sounding capability provided by FIR observations.

At ground level the atmosphere in the FIR spectral region is very opaque because of the high concentration of water vapour. For this reason, sounding of the FIR with ground-based observations is possible only from high altitude sites where the water vapour content is sufficiently low to make the atmosphere transparent at frequencies below 667 cm$^{-1}$. This requires the deployment of FIR instrumentation on high mountain sites, around 3000 m AMSL or at higher altitude (Palchetti et al., 2015; Mlynczak et al., 2016; Turner et al., 2012). Observations during the wintertime are preferred in order to have the water vapour content at its seasonal minimum, thus improving the capability to sound the lowest wavenumbers, with potential semi-transparency down to 250 cm$^{-1}$. The sounding of the remainder of the FIR, below 250 cm$^{-1}$, is only possible from airborne platforms, high-altitude aircraft (Cox et al., 2007) or balloons (Mlynczak et al., 2006; Palchetti et al., 2006), flying in the upper troposphere or the lower stratosphere.

In this paper, we describe the measurements acquired during the first field deployment of FIRMOS, between the end of 2018 and the beginning of 2019 at the Alpine observatory of Mt. Zugspitze in the south of Germany at about 3000 m altitude, together with all the ancillary measurements needed to provide a complete characterisation of the observed atmospheric and surface states.

## 2 Zugspitze Observatory

The Zugspitze site was chosen because of the very dry atmospheric conditions frequently encountered at the Summit Observatory, with values of the integrated water vapour (IWV) as low as 0.1 mm and median of 2.3 mm (see more details in Sussmann

et al., 2016). The minimum IWV is approximately a factor of 40 lower than at typical lowland mid-latitude sites. In this respect the Zugspitze is comparable to the driest sites in the world, i.e. remote locations like the Atacama desert or polar stations. In spite of that, the Zugspitze Observatory is reachable via cable car within 10 minutes from a car park and offered a suitable infrastructure for the deployment of FIRMOS due to the presence of a complementary and comprehensive set of instruments.

This provided independent information of the observed atmospheric and cloud states to be used as reference for the comparison with FIRMOS. The Zugspitze Observatory comprises two stations, the Summit station (47.421° N, 10.986° E, 2962 m AMSL) and the Environmental Research Station Schneefernerhaus (47.417° N, 10.980° E, 2675 m AMSL), which is located on the south slope of mountain, 300 m below and 680 m southwest from the Summit (see Fig. 1).

## 3 Instrument description and performed measurements

During the campaign, measurements were carried out using instruments deployed at the Zugspitze Summit, at the Schneefernerhaus station, and at the Karlsruhe Institute of Technology (KIT) in Garmisch-Partenkirchen.

The emission spectrum was measured in the FIR with FIRMOS, which was developed at CNR-INO with the support of ESA and the Italian Space Agency (ASI), and installed in November 2018 on the terrace of the Summit station. The instrument is a Fourier Transform Spectrometer (FTS) designed on the base of the heritage of a similar instrument, named Radiation

Explorer in the Far InfraRed – Prototype for Applications and Development (REFIR-PAD) (Bianchini et al., 2019), permanently installed in Antarctica since 2012 (Palchetti et al., 2015). The FTS consists in an interferometric layout with double-input and double-output ports in a Mach-Zehnder configuration, which increases the measurement reliability and the calibration precision (Carli et al., 1999). The instrument is able to cover the FIR radiation using wideband germanium-coated biaxially-oriented polyethylene terephthalate beam-splitters and room-temperature pyroelectric detectors. It was mounted within a plastic box to

protect optics and electronics from environmental conditions, such as wind and snowfall, when installed on the site. The full field-of-view (FOV) of FIRMOS is 22 mrad.

At the Summit station, KIT operates an Extended-range Atmospheric Emitted Radiance Interferometer (E-AERI) system (Sussmann et al., 2016), manufactured by ABB Bomem Inc. (Quebec, Canada), which covers part of the far, mid and near infrared spectral regions. As FIRMOS, E-AERI is zenith looking, its FOV is 46 mrad.

The spectrum of the Downwelling Longwave Radiation (DLR) was therefore measured in the FIR and mid-infrared spectral ranges from 100 to 1800 cm$^{-1}$ (100–5.56 $\mu$m) using both FIRMOS, covering the 100–1000 cm$^{-1}$ spectral range, and the first channel of E-AERI, covering the 400–1800 cm$^{-1}$ range. Spectra are provided without apodisation, with a sampling resolution corresponding to the sinc response function, of $\Delta\sigma = 0.3$ cm$^{-1}$ for FIRMOS (maximum optical path difference (MOPD) = 1.66667 cm) and $\Delta\sigma = 0.48215$ cm$^{-1}$ for E-AERI (MOPD = 1.03703 cm). The instrument line shape (ILS) of FIRMOS is

a linear combination of sinc and sinc$^2$ (as in the case shown in Bianchini et al. (2019)) with a full width at half maximum (FWHM) of about 0.36 cm$^{-1}$; whereas for E-AERI the ILS is an "ideal" sinc with FWHM = 0.58 cm$^{-1}$ (Knuteson et al., 2004). More information about ILSs is provided with the dataset.

Figure 2 shows FIRMOS installation close to the E-AERI instrument, which is permanently installed on the top of the shelter, 4 m above FIRMOS, and routinely performs measurements of spectral DLR when the weather conditions allow operation. A comparison between the average of 310 measurements of the spectral DLR in clear sky conditions performed simultaneously with FIRMOS and E-AERI is also shown in the figure.

KIT also operates two lidar systems for aerosols/clouds (Vogelmann and Trickl, 2008), and water vapour and temperature (Klanner et al., 2020) profiling, installed at the Schneefernerhaus station. The advantage of mounting the lidar systems at Schneefernerhaus and the spectrometer at the Summit is that the onset of the measured lidar profiles is typically about 300 m above the laser, which coincides with the location of the spectrometers at the Summit. Cirrus cloud properties are detected with the stratospheric aerosol lidar, which is a pure backscatter lidar operating at the wavelength of 532 nm. This system was operated in a semi-automatic mode, with a sequence of one profile every 4 or 10 minutes and an integration time of 1 min. Water vapour and temperature profiles were measured alternatively with a Raman lidar system using a XeCl-Laser at 308 nm for water vapour and with a Nd:YAG laser at 355 nm for temperature. The latter measurements were carried out manually in order to switch between one laser and the other. Water vapour profiles were recorded with 1 million laser shots, an integration time of 1 hour and a vertical resolution of 30 m near ground (4 km AMSL) and 250 m at 15 km AMSL. Temperature profiles were derived from density profiles measured only in absolute clear-sky conditions with an integration time of typically 1 hour and a vertical resolution of 30 m near ground (4 km AMSL) and 2 km at 80 km AMSL. The lidar in operation at night and an example of measurements are shown in Fig. 3.

Specific balloon radiosoundings (Klanner et al., 2020) for in-situ water vapour profiles and cirrus cloud microphysics were performed on two days (5–6 February 2019) under clear sky and thin cirrus cloud conditions during night time (Fig. 4). The antenna for communication with the balloon was built up on the Summit station and the preparation of the sonde payload and launch took place at KIT Campus Alpin, in Garmisch-Partenkirchen (47.476° N, 11.062° E, 730 m AMSL ) about 8 km northeast of Mt. Zugspitze. The balloon sondes are equipped with a standard radiosonde (Vaisala RS41-SGP), a Cryogenic Frostpoint Hygrometer (CFH) for measuring water vapour profiles with high accuracy, an ozone sonde, and the Compact Optical Backscatter Aerosol Detector (COBALD) (Brabec et al., 2012), developed at the Swiss Federal Institute of Technology in Zürich, that provides backscatter profiles at two wavelengths (940 nm and 455 nm). The colour index (CI) within the cloud layers, defined as the aerosol backscatter ratio (ABSR = backscatter ratio - 1) in the red channel (940 nm) divided by ABSR in the blue channel (455nm), is also provided. The combined balloon payload is well tested and also regularly used by the Global Climate Observing System (GCOS) Reference Upper Air Network (GRUAN) (see e.g. Dirksen et al., 2014). The CFH has an uncertainty of about 2–3% in the troposphere and less than 10 % in the lower stratosphere. The CFH is especially suitable for measuring water vapour under the dry conditions at the tropopause and in the stratosphere up to altitudes of 28 km. The accuracy of the backscatter ratio after post processing the COBALD raw data is better than 5 %, while the precision is smaller than 1 % in the upper troposphere / lower stratosphere (Vernier et al., 2015).

The Summit station is served with standard Pressure, Temperature, Relative Humidity (RH), and Wind (PTHW) meteo observations, provided by Deutscher Wetterdienst (DWD) at the website https://opendata.dwd.de/climate_environment/CDC/observations_germany/climate/, that for completeness were also included

in the available dataset for the period covered by the field campaign. Useful standard atmospheric soundings with operational radiosondes available near Zugspitze can be freely downloaded via the University of Wyoming website (http://weather.uwyo.edu/upperair/sounding.html) for the station of Muenchen-Oberschlssheim (Station No. 10868, daily soundings at 00 and 12 UTC), 100 km north the Summit, and the Innsbruck-Flughafen station (Station No. 11120, daily soundings at 03 UTC), 33 km southeast.

Finally, the properties of snow samples were characterised in terms of snow grain type, density ($kg\ m^{-3}$) and Specific Surface Area (SSA, $m^2\ kg^{-1}$), using the DUal Frequency Integrating Sphere for Snow SSA measurement (DUFISSS) sensor, which retrieves the SSA by measuring the reflectance at 1310 nm (Gallet et al., 2009).

## 4  Field campaign

The aim of the Zugspitze field campaign was to collect FIR measurements of the DLR to be used to provide evidence of the FIR capability to retrieve vertical profiles of water vapour, temperature and cirrus cloud properties, as well as to perform a side-by-side validation of the FIRMOS measurements in the spectral range in common with the E-AERI instrument permanently installed on the site.

The campaign took place from 29 November to 18 December 2018 and from 21 January to 20 February 2019. Measurements were performed when the weather conditions allowed operations; this occurred for a total of 33 days. The main specifications of the measurements, which are provided in the dataset, are summarised in Table 1.

During the first part of the campaign in November-December 2018, FIRMOS was installed and made operational. The weather conditions during that period were not very good and only a few days of clear sky to test acquisitions occurred (only 43 useful sky observations). This first part is considered an engineering campaign for the deployment and preparation of the full suite of instruments. For completeness, these measurements were included in the dataset but their usefulness is quite limited.

The second part of the campaign in January-February 2019 was dedicated to the acquisition of the DLR spectra,both in clear sky and in cirrus cloud conditions, and of all the ancillary information provided by lidars, radiosoundings, and meteo stations. Lidar measurements can be used to retrieve cirrus properties such as cloud geometry (cloud bottom and top heights), extinction profile, and optical depths. These properties, together with vertical soundings of humidity and temperature, provide a full characterisation of the atmospheric state that can be used to check and to refine radiative models of water vapour spectroscopy and cirrus ice-particle properties in the FIR and to explore the effect on the retrieval performance of cirrus micro-physics, as shown in Di Natale et al. (2020). An example of spectral measurement in the presence of cirrus cloud and the associated lidar profile is shown in Fig. 5.

Four days at the end of the campaign were finally devoted to measure snow and ice samples with the objective to characterize the FIR spectral emissivity, which is not well-characterised in the FIR range (Huang et al., 2016). We performed systematic snow measurements on 18 and 19 February 2019, FIRMOS was adapted to observe in a slant direction close to nadir (Fig. 6, top-left). In this way the main contributions to the measured radiance were by the surface emission and by the sky reflection. Each sample was measured by FIRMOS for roughly 90 minutes and successively characterised in terms of its surface

properties. An example of FIRMOS measurement with error estimates is shown in Fig. 6, top-right. To attain diversity in the snow properties, 8 samples of snow and 1 sample of ice were collected in the vicinity of the Zugspitze Summit and close to Zugspitzplatt station and characterised with DUFISSS (Fig. 6, bottom-left). One measurement with artificial patterns of surface roughness was also conducted. Figure 6 (bottom-right) shows the density and SSA values retrieved on the measured samples.

For the ice sample, the figure shows only a rough estimate of the parameters (grey box), since a precise measurement could not be performed with the available equipment.

## 5 FIRMOS validation

The FIRMOS dataset includes calibrated spectra and noise estimates, calculated starting from the FTS acquired interferograms and following the procedure described in Bianchini and Palchetti (2008). Each spectrum is the average of 4 sky observations and

160 it is calibrated with 4 calibration measurements (2 looking at a hot backbody reference source and 2 looking at a cold source). The noise is characterised by a spectrally-uncorrelated component (independent from one spectral channel to the other), called noise equivalent spectral radiance (NESR), due to the detector error, and a spectrally-correlated component, called calibration error (CalErr), mainly due to the temperature uncertainty of the calibration sources. The NESR is obtained from the error propagation through the calibration procedure of the $1\sigma$ detector random error, whereas the CalErr is calculated through the

165 error propagation of the temperature accuracy error measured on each reference blackbody (Bianchini and Palchetti, 2008). These noise estimates show highly resolved structures which are due to the absorption lines of gases inside the interferometric path. They depend on the actual working conditions of the instrument that can be vary from measurement to measurement. An example of the FIRMOS products is shown in Figure 7. In the figure, the NESR (red line) is also compared with the standard deviation (STD) of 4 sky measurements (brown line) measured with constant sky conditions. Compared to the NESR, the

170 STD estimate does not contain the noise coming from the calibration function but it contains the effect of possible radiance variations coming from the observed scene. Therefore the STD can be smaller then the NESR estimate when the observed scene is constant.

FIRMOS spectral measurements were validated against E-AERI in the common spectral region in order to qualify FIRMOS with a standard commercial spectrometer. The noise and calibration accuracy for E-AERI were calculated, using the standard

procedure provided by the manufacturer and outlined in Knuteson et al. (2004), from the variance of measurements observing constant sources and taking the average over 25 cm$^{-1}$ wavenumber bins across the spectrum. To compare these features with the analogous features of FIRMOS, the same 25 cm$^{-1}$ average was also operated. The comparison is shown in Fig. 8 in brightness temperature error at 280 K. The FIRMOS noise equivalent delta temperature (NEDT) was calculated from the STD, using a similar approach as for the E-AERI noise estimate. More information about the spectral and radiance calibration of the

two instruments are provided with the dataset.

A measurable error of the spectral frequency scale, due to the uncertainty on the metrology laser, was found on both FIRMOS and E-AERI by fitting the measurements with synthetic spectra. FIRMOS shows a frequency scale error of about +50 ppm between simulations and observations ($\sigma_{real} = (1+5\times10^{-5})\times\sigma_{firmos}$), whereas E-AERI has the same error but with different

sign (i.e. $\sigma_{real} = (1-5\times10^{-5})\times\sigma_{e-aeri}$). The frequency scale of spectra was therefore recalibrated with this correction before the comparison.

Furthermore, since the two instruments have different instrument line shapes and measurements are provided on different sampling grids, for this comparison spectra are equalised by applying the same apodization function (Norton-Beer strong) (Norton and Beer, 1976, 1977) with 0.968 cm$^{-1}$ of FWHM resolution and resampled to a common spectral grid of 0.5 cm$^{-1}$. Figure 9 shows the comparison between the mean spectra of 420 simultaneous measurements selected with the temporal distance between FIRMOS and E-AERI measurements less than 10 minutes. A good agreement between the two instruments is found in the common spectral range from 400 to 1000 cm$^{-1}$ with the difference (shown in the bottom panel) that is within the total uncertainty calculated using both the noise and the calibration accuracy (summed in quadrature) of both instruments. The difference is larger in the strong water vapour and $CO_2$ bands because the slightly different location of the two instruments. However, only in few lines in the range of 400–450 cm$^{-1}$ the difference between spectra exceeds the uncertainty estimate; this is probably due to the different FOV of the two instruments, which on average might observe a slightly different scene.

To make another assessment of FIRMOS measurements, an indirect approach was also used by comparing values of IWV retrieved from FIRMOS spectral measurements with the same parameter retrieved from E-AERI measurements, using in both cases only the common 400-600 cm$^{-1}$ spectral region. The IWV was calculated using the KIT algorithm that minimises the FIRMOS (or E-AERI) vs. LBLRTM (Line-By-Line Radiative Transfer Model) spectral residuals. While details will be the subject of an upcoming publication, we briefly state here that we found only a small overall bias (FIRMOS-AERI) = 0.0002 mm which is negligible compared to the measured atmospheric IWV states, ranging from 0.2 to 2 mm. This proved that there are no indications of significant calibration errors between the two instruments in the FIR spectral domain of the water vapour rotational band.

## 6 Conclusions

In summary, the unique spectral measurements provided by the combination of FIRMOS and E-AERI covering the relevant spectral region of the thermal emission of the atmosphere from 100 to 1800 cm$^{-1}$, together with the other supporting measurements (lidars, radiosoundings, atmospheric state, and surface properties), provide a complete dataset that can be used to constrain radiative properties of water vapour, cirrus ice particles, and snow/ice emissivity over almost all the infrared emission, including the under-explored FIR spectral range.

## 7 Data availability

The full dataset of the 2-month campaign, including infrared spectra (FIRMOS and E-AERI) and all the additional information (lidars, local PTHW, dedicated RS, snow SSA), is available via the ESA campaign dataset website https://earth.esa.int/eogateway/campaigns/firmos (Palchetti et al., 2020a, https://doi.org/10.5270/ESA-38034ee). ESA requires

a free registration to inform users about issues concerning data quality and news on reprocessing. Information about the data
formats are reported in README files within each data subdirectory.

*Author contributions.* LP designed the experiment and was chief scientist for the field campaign. RS was the responsible for the local deployment. LP, MB, GB, FDA, AM, SV, RS, MR, HV, CR, DK, BC run the instruments during the campaign. LP, CB, GDN, RS, MR, HV, CR, BC, SDB, MG, FGW carried out data analysis and measurement validation. DS and HO were responsible for the campaign organisation. LP prepared the manuscript with contributions from all co-authors.

*Competing interests.* The authors declare that they have no conflict of interest.

*Acknowledgements.* The authors acknowledge the European Space Agency (ESA) with the FIRMOS project (ESA–ESTEC Contract No. 4000123691/18/NL/LF) for the support to the development and the deployment of FIRMOS and the Italian Space Agency (ASI) with the research projects SCIEF (Italian acronym of Development of the National Competences for the FORUM experiment - ASI contract No. 2016-010-U.0) for the preliminary development of a few subsystems. The dedicated balloon activities were also partly supported by funding from the Helmholtz Association in the framework of MOSES (Modular Observation Solutions for Earth Systems).

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

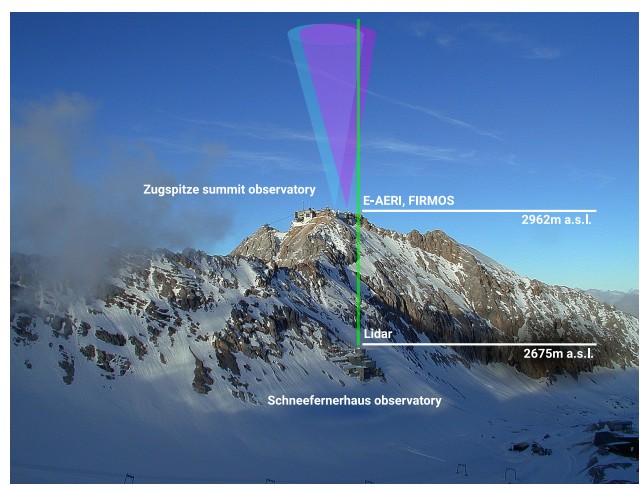

**Figure 1.** Summit and Schneefernerhaus stations at the Mt. Zugspitze, Germany.

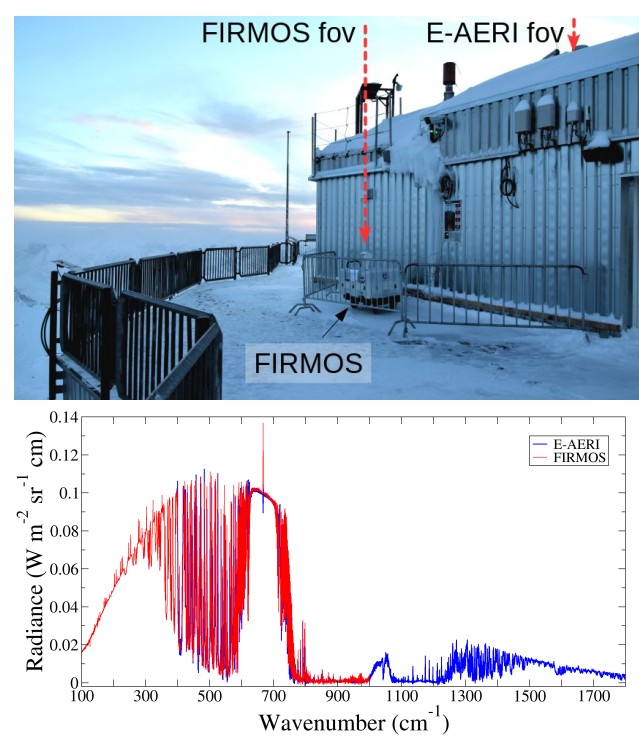

**Figure 2.** FIRMOS and E-AERI installed at the Summit station. The comparison between the average spectra of 310 simultaneous observations acquired by the two instruments in clear sky during 2019 campaign is shown in the bottom panel.

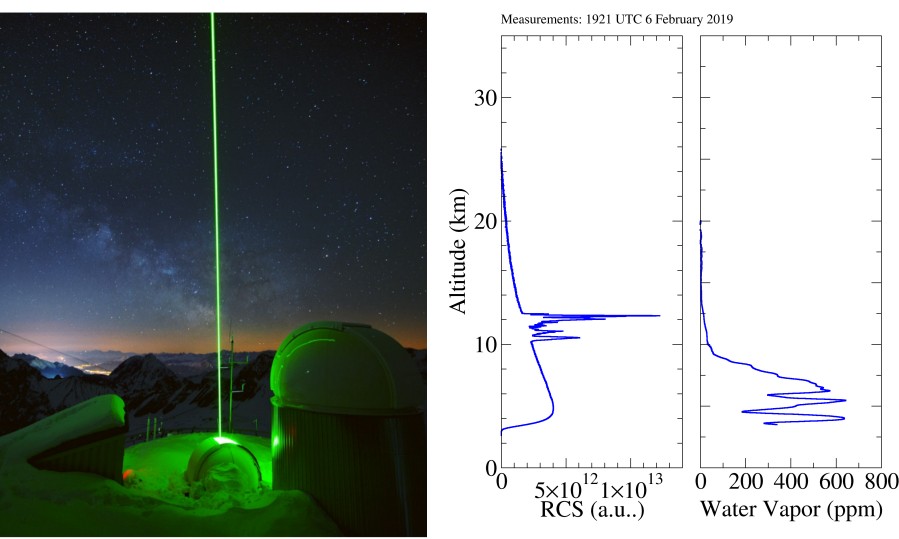

**Figure 3.** Lidar measurements at the Schneefernerhaus station. The plot shows an example of the Range Corrected Signal (RCS) of the backscatter lidar, detecting a cloud layer, and the water vapour profile of the Raman lidar measured on 6 February 2019, 19:21 UTC.

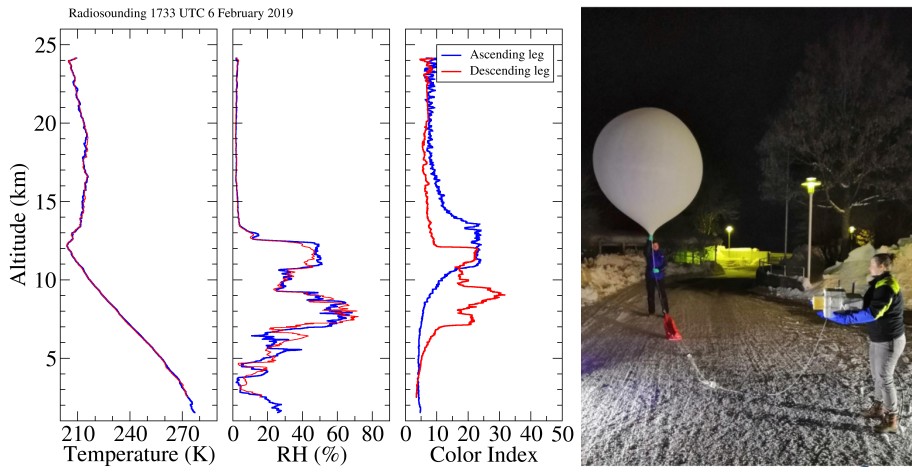

**Figure 4.** Radiosoundings from Garmisch-Partenkirchen of temperature, relative humidity (RH), and colour index profiles for the launch on 6 February 2019, 17:33 UTC. A single cloud layer during the ascending leg (blue curve) and a double layer during descending (red curve) is shown in the colour index plot.

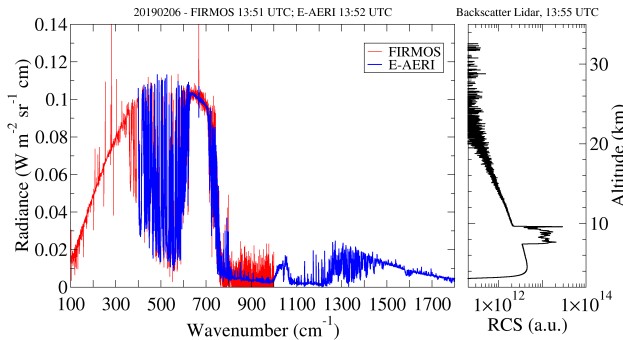

**Figure 5.** FIRMOS and E-AERI spectra simultaneously acquired in presence of a cirrus cloud on 6 February 2019 and the corresponding lidar RCS profile.

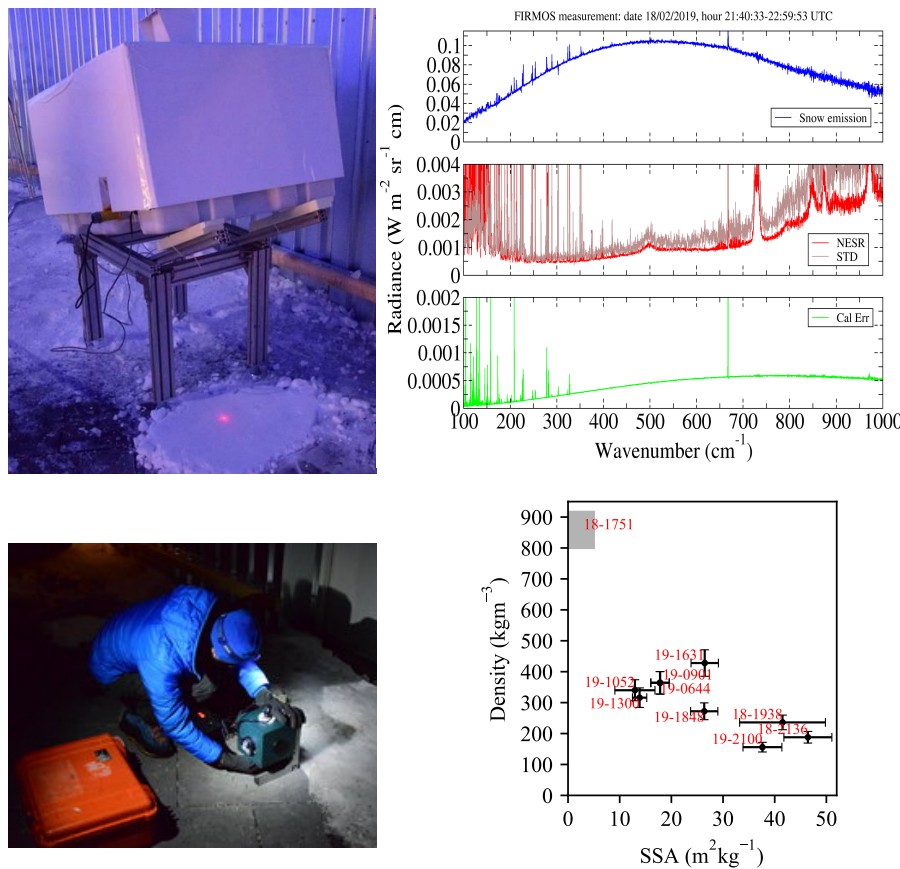

**Figure 6.** Snow measurements at Zugspitze Summit. FIRMOS in a slant angle during snow emissivity measurement (top-left) and the average spectrum with uncertainties of 11 spectra acquired measuring the 18-2136 sample (top-right). DUFISSS snow Specific Surface Area (SSA) measurement (bottom-left) and density-SSA properties of the snow samples (bottom-right, day-time of FIRMOS acquisition labelled in red). The upper-left grey box corresponds to an ice sample whose parameters were only roughly estimated.

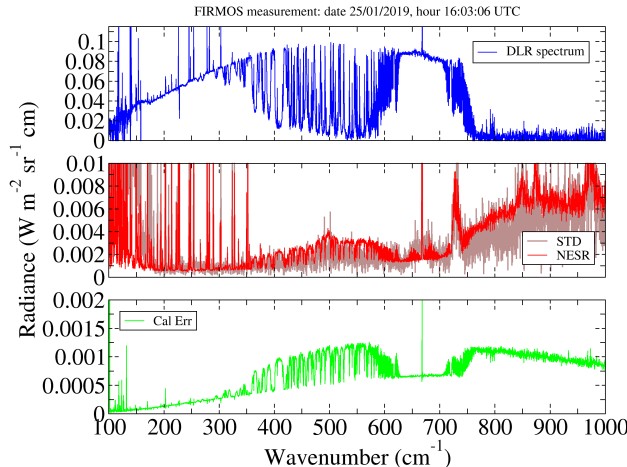

**Figure 7.** Example of FIRMOS DLR spectrum and the associated noise estimates acquired on 25 January 2019, 16:03 UTC.

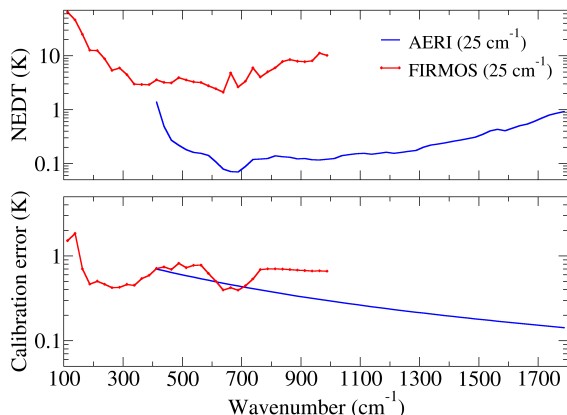

**Figure 8.** FIRMOS (red lines) and E-AERI (blue lines) brightness temperature error estimates at 280 K, averaged over 25 cm$^{-1}$ wavenumber bins across the spectrum.

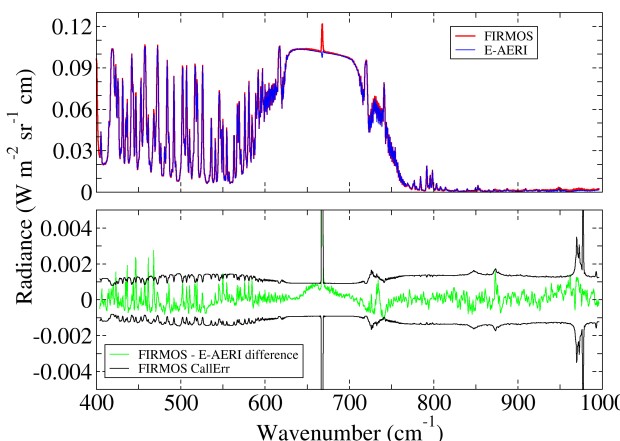

**Figure 9.** Comparison between FIRMOS and E-AERI averaged spectra of 420 simultaneous acquisitions for the 2019 campaign. Spectra were apodised with the Norton-Beer strong with FWHM=0.968 $cm^{-1}$ and resampled to a common spectral grid of 0.5 $cm^{-1}$. The bottom graph shows the difference between the average spectra (in green), compared with the total measurement uncertainty (in black).

**Table 1.** Instrumentation deployed at Zugspitze and measurements performed during the FIRMOS campaign and provide in the available dataset.

| Instrument | Location | Type of measurement | Integration / repetition time | Date / No of measurements |
|---|---|---|---|---|
| FIRMOS | Zugspitze Summit station | DLR spectrum FOV = 22 mrad 100-1000 cm$^{-1}$ $\Delta\sigma = 0.4$ cm$^{-1}$ | 128 s / 256 s | 29/11/2018 to 18/12/2018 1197 spectra |
| | | $\Delta\sigma = 0.3$ cm$^{-1}$ | 210 s / 420 s | 21/1/2019 to 15/2/2019 838 spectra |
| FIRMOS | Zugspitze Summit station | Ice/snow spectrum FOV = 22 mrad 100–1000 cm$^{-1}$ $\Delta\sigma = 0.3$ cm$^{-1}$ | 210 s / 420 s | 16/2/2019 to 20/2/2019 152 snow + 283 sky spectra |
| E-AERI | Zugspitze Summit station | DLR spectrum FOV = 46 mrad 400-1800 cm$^{-1}$ $\Delta\sigma = 0.48215$ cm$^{-1}$ | 214 s / 440 s | 28/11/2018 – 20/2/2019 4987 spectra |
| Meteo station from DWD | Zugspitze Summit station | Local Pressure, Temperature, Relative Humidity, and Wind | every 10 min | Continuous |
| Backscatter lidar | Schneefernerhaus station | Extinction profile | 1 min / 4-10 min | 16/11/2018 to 19/02/2019 |
| Raman lidar | Schneefernerhaus station | H$_2$O and T profiles | 1 h | 16/11/2018 to 19/02/2019 |
| Dedicated RS | IMK-IFU, Garmisch-Partenkirchen | Pressure, Temperature, Relative Humidity, O$_3$, cloud extinction and colour index profiles | - | 05/02/2019 – 06/02/2019 4 launches under clear sky and thin cirrus cloud conditions during night time |
| DUFISSS | Zugspitze Summit station | Snow Specific Surface Area | - | 18/02/2019 – 19/02/2019 2 measurements every FIRMOS acquisitions |