# Peer review of "Observations of the downwelling far-infrared atmospheric emission at the Zugspitze observatory"

_Earth System Science Data, 2020_

## Author Comment (AC1)

**Reply to Referee comment essd-2020-377-RC1**

We thank the reviewer for the positive comments. Here below the reply to the specific minor comments.

1) In the revised text, we have added the required information:
- The field-of-views (full angle) of the spectrometers have been added both in the text and in Tab.1:
  - FIRMOS FOV = 22 mrad
  - E-AERI FOV = 46 mrad
- A new figure, shown here below, will be added in the revised paper to show the comparison both for the mean NeDT and the calibration accuracy in brightness temperature units calculated at 280 K for the whole dataset. For this comparison, an average over 25 cm$^{-1}$ wavenumber bins across the spectrum, which is used for E-AERI, is also applied to FIRMOS estimates.

[Figure]

2) Actually, the frequency scale residual difference between the two instruments was corrected before the comparison.
The instrument FOV of the interferometer is responsible for part of this effect (30 ppm for FIRMOS and 132 ppm for E-AERI); another contribution comes from the calibration of the metrology system. However, all these contributions are corrected globally during the spectral calibration in both instruments. Nevertheless, a residual small difference of -50 ppm for E-AERI and + 50 ppm for FIRMOS was found with an a-posteriori fit of the frequency scale. Therefore before the comparison, shown in Fig. 7, these correction factors were applied with the Norton-Beer apodization and the resampling on the same spectral grid. We notice also that both instruments have an instrument line shape very close to the sinc function (see also the reply to the review RC3), which is further equalised by the Norton-Beer apodization before the comparison.

For a better estimate of the residual differences, Figure 7 will be updated with the figure here below, where we have considered the total uncertainty calculated using both the noise and the calibration accuracy (summed in quadrature) of both instruments.

[Figure]

We have also investigated the effect of the different observing locations of the two instruments. As described in the paper, E-AERI is located on the roof-top of the Zugspitze observatory, whereas FIRMOS is installed on the terrace below, 4 m lower, in a southward corner protected from winds. In these conditions, the different mean temperature and humidity close to the instruments, and the presence of an additional layer of 4 meters of air in front of FIRMOS could produce some differences in the lines of $CO_2$ and $H_2O$ but always below the calibration error estimated for FIRMOS, as shown by the sensitivity study in the figure here below.

[Figure]

The residual differences still present in the range 400-450 $cm^{-1}$ might be due to the different instrument FOVs, which on average might observe a slightly different scene.

3) The acronyms in Table 1 have been written in full names.

---

## Author Comment (AC2)

**Reply to Referee comment essd-2020-377-RC2**

We thank the reviewer for the positive comments and the suggested minor corrections, which are all implemented in the revised text.

Concerning the comment on "how the dataset can be used to constrain radiative properties of ... cirrus ice particles, and snow/ice emissivity", we will improve the text in the revised paper by adding the following information.

Lidar measurements can be used to retrieve cirrus properties such as cloud geometry (cloud bottom and top heights), extinction profile, and optical depth which provide, together with vertical soundings of humidity and temperature, a full characterisation of the atmospheric state that can be used to check and refine radiative models of water vapour spectroscopy and cirrus ice-particle properties in the FIR and to explore the effect on the retrieval performance of cirrus micro-physics as shown in Di Natale et al., 2020, DOI: 10.3390/rs12213574.

An example of spectral measurements in presence of the cirrus cloud, as shown in figure below, will be added to the revised text.

[Figure]

Comparison of FIRMOS and E-AERI spectra acquired in presence of a cirrus cloud on 6 February 2019 at 13:51 UTC and the corresponding lidar RCS profile.

Furthermore, the example of snow measurement of the figure below will also be added as an additional panel in Fig. 5.

[Figure]

Concerning the comment on page 75, only the first channel of E-AERI is used for this work. The value of 3.3 microns refers to the second channel of E-AERI and was erroneously used in the text instead of the corrected valued of 5.56 micron (1800 cm$^{-1}$) of the first channel. This mistake will be corrected in the revised paper.

Finally, all the suggested minor corrections will be done in the revised paper.

---

## Author Comment (AC3)

**Reply to Referee comment essd-2020-377-RC3**

We thank the reviewer for the positive comments and the highlighted issues, which allowed us to improve the final version of the paper and the dataset. Here below a detailed discussion is provided.

**Reply to general comment on the dataset**

The file structure will be simplified and a new version of the dataset will be provided. The new FIRMOS dataset will be downloadable as a single file.

**Reply to specific comments**

- Line 76: The spectral resolution refers to the sampling resolution $\Delta\sigma$ of the sinc function, which is 0.3 cm$^{-1}$ for FIRMOS (corresponding to OPDmax = 1.66667 cm) and 0.48215 cm$^{-1}$ for E-AERI (corresponding to OPDmax = 1.03703 cm). The instrument line shape (ILS) for E-AERI is corrected to obtain an "ideal" sinc following the procedure outlined in Knuteson et al. 2004, DOI: 10.1175/JTECH-1663.1. The ILS for FIRMOS is characterised a-posteriori with a retrieval procedure and it is approximated by a linear combination of sinc and sinc$^2$ (for more details see Bianchini et al. 2019, DOI: 10.5194/amt-12-619-2019), i.e.

$$ILS(\sigma) = \alpha(\sigma) \cdot sinc\left(\frac{\sigma}{\Delta\sigma}\right) + (1-\alpha(\sigma)) \cdot sinc^2\left(\frac{\sigma}{2\Delta\sigma}\right)$$

with

$$\alpha(\sigma) = sinc\left(\frac{\sigma\Omega}{4\Delta\sigma}\right)$$

where $\sigma$ is the wavenumber and the parameter $\Omega$, fitted in the retrieval procedure, has an average value of 0.001 sr

The resolution FWHM is approximated 1.207*$\Delta\sigma$, i.e. equal to about 0.36 cm$^{-1}$ for FIRMOS and 0.58 cm$^{-1}$ for E-AERI. All these information will be added in the revised text and the README files

- Line 145: The narrow spectral features in the FIRMOS NESR come from the high-resolution calibration function (the instrument gain function) which contains the absorption features of gasses inside the interferometric path. In the FIR and MIR spectral regions at ground level, water vapour and carbon dioxide absorption inside the instrument has an important contribution. Furthermore, the NESR estimate contains also the noise contribution coming from the measurement of the calibration function, which is performed every 4 sky measurements, and this contribution is comparable with the noise on the sky measurement. The standard deviation estimate is instead the variance of the 4 sky measurements, which are averaged in the final spectrum. It does not contain the noise coming from the calibration function but it contains the effect of possible radiance variations coming from the observed scene. Therefore the standard deviation can be smaller then the NESR estimate when the observed scene is constant. The procedure with which these errors are estimated is described in details in Bianchini and Palchetti 2008, DOI: 10.5194/acp-8-3817-2008.

- Line 154: The frequency scale factor is a constant parameter that can be easily corrected before the application.

- Line 158: Yes, that's correct, we refer to the FWHM of the applied Norton-Beer strong that is 0.968 cm$^{-1}$.

- Line 160: The a-posteriori bias correction was performed using the residual difference after a fit of the atmospheric state (retrieval analysis) in clear sky conditions and not by a comparison with E-AERI. The comparison with E-AERI was done only for the validation; thus the two measurements are completely independent. The note in the README.firmos will be corrected.

- Line 163: We have made a more accurate estimate of the total error to be applied to the comparison, including all the sources from both instruments, and now the residual differences between the two measurements are only present in a few lines in the 400-450 cm$^{-1}$ region probably due to slightly different observed scenes because the different FOVs, see also the reply to question 2 of RC1. An improvement of phase error correction algorithm, which might improve the calibration accuracy above 700 cm$^{-1}$, will be evaluated in next studies.

- Figure 2, here the spectra are on different sampling grid, thus it is not possible to show the residual difference.

- Figure 5: It will be updated using the figure below

[Figure]

- Table 1: Dates for DUFISSS measurements (18th and 19th February 2019) will be added in Table 1.

- Snow data: We now provide a more comprehensive description of the data in a new README.ssa_rho as well as contact information if any further questions arise.
The data used in Fig. 5 are written in the file SSA_rho.csv (for the dots, corresponding to the measured values) and in the file mV_g.csv for the error bars (qmV and qg columns, 0 denoting a good measurement ($\pm$10% of the measured value), 1 a medium quality measurement ($\pm$15%), and 2 a poor quality measurement ($\pm$20%)). SSA and density are computed from the raw measurements in mv_g.csv (with calibrations from calibs.csv), using the retrieval process described in Gallet et al. 2009, DOI: 10.5194/tc-3-167-2009.
Concerning the sample 18-1751, it is an ice sample used as a reference body, but differs from the snow samples: it is a limit case in terms of density (highest possible) and SSA (lowest possible). This is what drove our interest, though it was not possible to characterize its properties as finely as for the snow samples. Indeed, ice density cannot be measured with a snow sampler (the material is too hard). As a matter of fact, we made a rough estimate of the sample density as ice was not the main matter of the snow emissivity study. While pure Ice density is well known (917 kg.m$^{-3}$), our sample contained air bubbles lowering its density to an estimated 850kg.m$^{-3}$ (a typical value in the case of glacier ice) +/15%, which was very conservative. However we agree that this treatment lacked rigour. Reconsidering it, we propose to take a fixed range 800-917 kg m$^{-3}$. Similarly, we couldn't "measure" the SSA of ice with DUFISSS (too hard for the sampler, and too shallow

sample). Such measurement wouldn't make much sense as pure ice is a limit case for the definition of SSA (SSA is defined as the area of air/ice interface per kilogram of snow, in $m^2\ kg^{-1}$, so pure ICE would have an SSA of 0), and for the optical theory behind SSA measurements (Gallet et al., 2009). However, in the shortwave spectrum, pure ice is similar to a snow with an SSA of 3 $m^2\ kg^{-1}$ (Quentin Libois, François Tuzet, personal communication), this is why we put this value in the graph. Finally, we can fix a higher bound for the SSA of glacier ice at 5 $m^2\ kg^{-1}$, which is a lower boundary for dense snow (see e.g. Fig 10 of Tuzet et al. 2019, DOI: 10.5194/tc-13-2169-2019).

We acknowledge the value we used for SSA and the way it was presented in Fig. 5 lacked rigour and might be misleading. We propose to replace the value of 3 $m^2\ kg^{-1}$ by a value range 0-5 $m^2\ kg^{-1}$ for the SSA of the ice sample, bounded by the value for pure ice (0 $m^2\ kg^{-1}$) and by the lowest measured values for snow (5 $m^2\ kg^{-1}$).

In agreement with these corrections, Fig.5 was updated with a grey box for the ice sample corresponding to the updated bounds. This layout evidences that only a rough estimate of the sample parameters was made.

**Reply to technical corrections**

All the suggested technical corrections will be accepted

---

## Author Response (AR2)

**Reply to Referee comments of the Revised Submission – essd-2020-377**

**Referee #1 Comment:**

I appreciate the effort that the authors have made to answer the comments I raised. My concerns have been well addressed and I have no further comments. I recommend acceptance after one minor typo being fixed. The typo is on Line 79, it should be "1.66667 cm" instead of "cm-1" since this is MOPD.

**Reply**
The typo was corrected in the revised version

**Referee #2 Comment:**

This paper is basically ready for publications. Here are some suggestions for minor revision:
line 21 – "the space" should be "space"
77 – No comma after "resolution".
84 – Delete "the" before "operation".
183 – Should be "shapes".
184 – Should be "grids".
196 – "will be the subject"

**Reply**
All these corrections were made in the revised version.

**Referee #3 Comment:**

Lines 161-169 of the revised manuscript: I am still puzzled about the strong high resolution signatures in the NESR. To my understanding this is an artefact of the NESR determination rather than the real NESR of the instrument. I do not want to start a lengthy discussion about this point but I would like to see a comment in the text that the highly resolved structures in the NESR are due to gases inside the interferometric path.

**Reply**
This sentence was added to the revised version:
"These noise estimates show highly resolved structures which are due to the absorption lines of gases inside the interferometric path. They depend on the actual working conditions of the instrument that can be vary from measurement to measurement. "